# Abattoir Countrywide Survey of Dairy Small Ruminants’ Haemonchosis in Greece and Associated Risk Factors

**DOI:** 10.3390/ani15040487

**Published:** 2025-02-09

**Authors:** Konstantinos V. Arsenopoulos, Athanasios I. Gelasakis, Elias Papadopoulos

**Affiliations:** 1Department of Veterinary Medicine, School of Veterinary Medicine, University of Nicosia, Engomi, 2414 Nicosia, Cyprus; 2Laboratory of Anatomy and Physiology of Farm Animals, Department of Animal Science, Agricultural University of Athens (AUA), Iera Odos 75 Str., 11855 Athens, Greece; gelasakis@aua.gr; 3Laboratory of Parasitology and Parasitic Diseases, School of Veterinary Medicine, Faculty of Health Sciences, Aristotle University of Thessaloniki, 54124 Thessaloniki, Greece; eliaspap@vet.auth.gr

**Keywords:** *Haemonchus contortus*, sheep, goats, epizootiology, risk factors, prevalence

## Abstract

This study investigated the prevalence of *Haemonchus contortus* infections in dairy sheep and goats in Greece and evaluated host-related risk factors. A total of 1004 abomasa were examined and a structured questionnaire collected farm and animal data. Molecular identification of *Haemonchus* spp. revealed a 37.2% prevalence of mono-species *H. contortus* infections. In sheep, key preventive factors included anthelmintic treatments, young age, and intensive management system. In goats, intensive management, anthelmintic treatments, high-altitude farms, and spring/summer seasons were significant in reducing infection risk.

## 1. Introduction

Gastrointestinal nematodes (GINs) threaten the sustainability of ruminant livestock production on farms worldwide [1,2]. Epizootiological studies have revealed their presence among domestic ruminants in central and northern European countries [3,4,5], as well as in Mediterranean countries [6,7,8], including Greece [9,10]. *Haemonchus* spp. are listed among the most commonly detected GIN [11,12] with severe implications on ruminant production, including heavy losses in terms of milk and meat production (quantitatively and qualitatively), reduced offspring growth, impaired health, welfare, nutritional status, inefficient reproduction, and increased costs for prevention and treatment, as well as predisposition to other diseases [12,13,14].

In Greece, GIN infections are particularly widespread among small ruminants [9,10], with several studies reporting *Teladorsagia* spp. along with *Haemonchus* spp. to be the most prevalent parasites in dairy sheep and goats [15,16]. This is mainly attributed to the predominant semi-intensive farming system, where the overall feeding scheme includes a combination of grazing in natural pasturelands and the supplementary feeding of concentrates and roughages, year-round [17]. Seasonal variations play a crucial role, as environmental conditions during hot and humid months favor the development and survival of infective larvae. Host factors, such as age and sex, contribute to susceptibility, with young female animals, particularly during lactation, being more affected. Additionally, altitude influences parasite transmission, as lowland regions typically present higher parasite burdens than high-altitude areas. The co-existence of sheep and goats may further exacerbate transmission dynamics, while the widespread emergence of anthelmintic resistance complicates effective control measures [10,14].

Despite the documented significance of haemonchosis for the dairy small ruminant sector in Greece, there is a scarcity of updated epizootiological data at a countrywide level. Greek islands play a significant role in the epizootiology of *Haemonchus contortus* (Rudolphi, 1803), which can be justified based on several ecological, geographical, and livestock management factors. Moreover, all of the available studies were based on a coprological examination by microscopy of L_3_ larvae for the diagnosis of infections, which is less sensitive for identifying nematodes at the species level.

Therefore, the objectives of the present study were (a) to determine the prevalence of haemonchosis in dairy sheep and goats, based on the post-mortem examination in abattoirs of both continental and insular Greece, and (b) to link it with potential host-related traits such as the age and the sex, and farm-related traits such as the altitude of the farm, the farming system, the co-existence of goats and sheep in the same flock, the season, and the type of anthelmintic treatment used on the occurrence of haemonchosis.

## 2. Materials and Methods

### 2.1. Study Design—Methodology

An extensive countrywide cross-sectional study was conducted between 2018 and 2020, to determine the epizootiology of haemonchosis in small ruminants in Greece. This study was based on the post-mortem examination of sheep and goats’ abomasa collected from 67 abattoirs across the country (i.e., 48 from continental and 19 from insular country) (Figure 1). Both sheep’s and goats’ abomasa were collected from each abattoir visited around Greece.

In total, 1004 abomasa collected from 573 (57.1%) sheep and 431 (42.9%) goats, originating from 306 and 300 flocks, respectively, were examined for the presence of the nematode *H. contortus*.

### 2.2. Collection and Post-Mortem Examination of Sheep’s and Goats’ Abomasa from Abattoirs

During the evisceration of the gastrointestinal tract, each abomasum was separated, opened with scissors, and its content removed; then, it was individually placed in isothermal plastic bags within 30 min post-evisceration to be transported under refrigeration (+4 °C) to the Laboratory of Parasitology and Parasitic Diseases. Upon arrival at the laboratory, a detailed examination of the mucosa of the abomasum was performed to detect adult *Haemonchus* helminths. A positive result for *Haemonchus* spp. infection was defined by the presence of, at least, one adult nematode parasite on the mucosa of the abomasum.

### 2.3. Data Collection

Along with the collection of abomasa from the abattoirs, relevant data (Table 1) were also collected with regard to the location of the abattoirs and the slaughter date, as well as the age and species of the slaughtered animal, the farming system, the co-existence of goats and sheep within the flock of origin, the anthelmintic treatment protocol, and the farm’s altitude. Farms were classified according to (i) the altitude (plain farms < 300 m and semi-mountainous or mountainous farms > 300 m), (ii) the farming system (grazing or zero-grazing), (iii) the application of anthelmintic treatment (a. pro/benzimidazoles, b. macrocyclic lactones, c. combined use of pro/benzimidazoles and macrocyclic lactones, and d. no anthelmintic treatment), and (iv) the co-existence of sheep and goats. Animals were grouped according to their species (sheep or goat), age-class (<2 months old, >2 to 15 months old, and >15 months old) and sex (male or female).

### 2.4. Data Handling—Statistical Analyses

Prevalence values and their 95% confidence intervals (CI 95%) were estimated using the Epitools (https://epitools.ausvet.com.au/ciproportion, accessed on 20 September 2024) and the Wilson score interval method. A binary logistic regression model was used to assess the contribution of potential risk factors on the *Haemonchus* spp. infection status in sheep and goats. Initially, a full factorial model was built and potential interactions were tested to check for significant confounders. No significant effects of these interactions were found; hence, a ‘main-effects’ model was used instead. The aforementioned risk factors included co-existence of other ruminant species, type of anthelmintic treatment, age, sex, farm altitude, farming system, and season, as described in Equation (1):Y = α + β_1_Χ_1_ + β_2_Χ_2_ + β_3_Χ_3_ + β_4_Χ_4_ + β_5_Χ_5_ + β_6_Χ_6_ + β_7_Χ_7_(1)
where Y = the probability of a small ruminant (sheep and goats were considered separately) being positive for *Haemonchus* spp., β_1_ to β_7_ are the regression coefficients of the coexistence of other ruminant species (X_1_, 2 levels: 0 = no other ruminant species, and 1 = at least one co-existent ruminant species), type of anthelmintic treatment (X_2_, 4 levels: 0 = pro/benzimidazoles, 1 = macrocyclic lactones, 2 = combination of pro/benzimidazoles and macrocyclic lactones, and 4 = no treatment), age class (X_3_, 3 levels: 0 ≤ 2 months 1 = 2 to 15 months, and 3 ≥ 15 months), sex (X_4_, 2 levels: 0 = male, and 1 = female), farm altitude (X_5_, 2 levels: 0 ≤ 300 m above sea level, and 1 ≥ 300 m above sea level), farming system (X_6_, 2 levels: 0 = semi-intensive, and 1 = intensive), and season (X_7_, 4 levels: 0 = spring, 1 = summer, 2 = autumn, and 3 = winter), respectively.

The Wald χ^2^ statistic of their regression coefficients (βs) was used to test the statistical significance of individual predictors. The Hosmer–Lemeshow (H–L) test, the Cox and Snell R^2^, and the Nagelkerke R^2^ indices were used to assess the goodness-of-fit for each individual model. Statistical significance was set at the 0.05 level.

### 2.5. Molecular Identification of Haemonchus contortus

All adult helminths, resembling *Haemonchus* spp., were collected from the abomasum of each animal and preserved in vials containing 99% ethanol, even though our primary goal was to obtain samples for molecular analysis rather than to assess the parasitic burden. For DNA extraction, 5595 adult *H. contortus* (15 helminths per infected abomasum) were included in our molecular approach. In the case where less than 15 adult parasites were detected in an abomasum, the respective number was replaced from *H. contortus* derived from farms that raised both sheep and goats. Then, the head of each helminth was carefully dissected at the cervical papillae, excluding the uterus and eggs. Genomic DNA was extracted following the protocol by Hillis et al. [18]. A 321 base pair fragment of the internal transcribed spacer 2 (ITS2) region of nuclear DNA (Figure 2) was amplified for species identification, using the primers NC1-forward: 5′-ACGTCTGGTTCAGGGTTGTT-3′ and NC2-reverse: 5′-TTAGTTTCTTTTCCTCCGCT-3′ described by Arsenopoulos et al. [19] and synthesized by Eurofins Genomics GmbH (Ebersberg, Germany).

## 3. Results

### 3.1. Prevalence of Haemonchus contortus Infection

The overall prevalence of *H. contortus* infections in the studied small ruminants was 37.2% (373/1004), with the infection rates for sheep and goats being 41.0% (235/573) and 32.0% (138/431), respectively. Μore details are presented in Table 2.

### 3.2. Molecular Identification of Haemonchus spp.

Helminths, resembling *Haemonchus* spp., were detected on the abomasum of all infected sheep and goats. The Blast analysis of sequences, obtained after molecular assessment of each helminth, showed a high nucleotide identity ranging from 99.9% to 100.0% with those of *H. contortus* available in the GenBank.

### 3.3. Descriptive Results

Table 3 summarizes the prevalence of *H. contortus* infections in sheep and goats for each level of the studied host- and environment-related risk factors. One-third of the studied sheep and goats were reared in mixed-species flocks. More than half of the animals received anthelmintic treatment with pro/benzimidazoles compared to other antiparasitic options, while approximately 20% of the animals received no anthelmintic treatment. Approximately 40% of goat kids < 2 months old were found parasitized by adult *H. contortus*. A lower percentage of dairy small ruminants were infected with *H. contortus* at altitudes > 300 m above sea level. Intensively managed (no grazing) sheep (17.8%) and goats (5.8%) suffered from haemonchosis. Female animals were infected with *Haemonchus* helminths at a higher percentage compared to male ones. Finally, a noteworthy percentage of sheep (23.4%) and goats’ (18.8%) abomasa were parasitized with adult *H. contortus* during winter.

### 3.4. Risk Factors of Sheep Infected by Haemonchus contortus

Table 4 summarizes the effects of the studied risk factors on the likelihood of *H. contortus* infections in sheep. Sheep treated with pro/benzimidazoles, macrocyclic lactones, or a combination of the two were 9.1 (95% CI: 4.2 to 20.0, *p* < 0.001), 7.7 (95% CI: 3.1 to 16.7, *p* < 0.001), and 8.3 (95% CI: 3.6 to 20, *p* < 0.001) times, respectively, less likely to be infected by *H. contortus* compared to non-treated ones. Likewise, <2-month-old lambs were 5.0 times less likely to be infected by *H. contortus* compared to >15-month-old sheep (95% CI: 2.9 to 8.3, *p* < 0.001). The likelihood of *H. contortus* infection was increased by 2.7 times in semi-intensively reared sheep compared to the intensively reared ones (95% CI 1.7 to 4.3, *p* < 0.05).

On the contrary, the co-existence of sheep and goats, the farm altitude, the sex of the slaughtered animals and the season of sampling did not have any significant (*p* > 0.05) effects on the likelihood of *H. contortus* infection in the studied animals.

### 3.5. Risk Factors of Goats Infected by Haemonchus contortus

Table 5 summarizes the effects of the studied risk factors on the likelihood of *H. contortus* infections in goats. Goats treated with pro/benzimidazoles, macrocyclic lactones, and a combination of the two were ca. 50 (95% CI: 10 to 250, *p* < 0.001), 100 (95% CI: 14 to 500, *p* < 0.001), and 100 (95% CI: 14 to 333, *p* < 0.001) times, respectively, less likely to be infected by *H. contortus* compared to non-treated ones. Goats reared at farms located at <300 m a.s.l. were ca. 2.6 times more likely to be infected by *H. contortus* compared to the ones at farms located at >300 m a.s.l. (95% CI: 1.4 to 4.8, *p* < 0.01). The likelihood of *H. contortus* infection was increased by 3.3 times in semi-intensively reared goats compared to the intensively reared ones (95% CI: 1.1 to 9.6, *p* < 0.05). Moreover, a statistically significant decrease in the likelihood of a goat to be found positive for *H. contortus* infection was observed in spring (by ca. 2.6 times, 95% CI: 1.3 to 5.3 times, *p* < 0.01) and summer (by ca. 2.6 times, 95% CI: 1.2 to 5.6 times, *p* < 0.05) compared to winter.

The co-existence of goats and sheep, age, and sex of the studied animals had no significant (*p* > 0.05) effects on the likelihood of *H. contortus* infections in goats.

## 4. Discussion

This is the first countrywide abattoir survey for the investigation of *H. contortus* infections in dairy small ruminants in Greece, which includes not only the continental but also the insular country, to the best of our knowledge. The Greek islands, due to their unique climate, geographic isolation, and traditional livestock management practices offer valuable conditions for studying the epizootiology of *H. contortus*. It is also the first time that host- and farm-related risk factors (i.e., age, sex, farm location altitude, farming system, the co-existence of sheep and goats in the same flock, season, and the anthelmintic treatment used) were assessed as potential risk factors of the *H. contortus* infections.

Epizootiological studies on haemonchosis are mainly focused on sequential observations of the worm burdens in grazing ruminants, searching for direct relationships between nematode infections and environmental factors. These types of studies are based on three different approaches [20] including (1) the use of animals—tracers, which graze on pasturelands contaminated with worm eggs at specific time-points, (2) worm counts from ruminants naturally infected and continuously challenged by nematodes, and (3) abattoir surveys [20], which is also the case in our study.

Abattoir surveys allow the direct estimation of *Haemonchus* spp. from grazing animals, providing precise data on the actual parasitic burden, while they can identify different life stages of this parasite, providing more comprehensive data on the life cycle and population dynamics of *Haemonchus* spp. On the contrary, studies based exclusively on faecal worm egg counts (FWECs) are not necessarily indicative of the real parasitic burden, which can be influenced by factors such as the egg production rates and faecal consistency. Another advantage of abattoir surveys is the detection of immature stages of parasites that have not yet started producing eggs, offering insights into early infection stages that FWECs cannot capture. Moreover, abattoir surveys provide direct evidence of infection, leading to more accurate estimates of the prevalence and intensity of infection, and can be integrated with other health surveillance activities conducted at slaughterhouses, such as monitoring for other diseases and contaminants, providing a holistic view of animal’s health. While abattoir surveys may have higher initial costs, they can be more cost-effective in the long run due to the large volume of data collected from animals already being processed for meat production. This can reduce the need for extensive on-farm sampling and testing [20].

In general, prospective epizootiological studies have provided a good understanding of the seasonal effects of *H. contortus* burdens for a range of environments, to set the basis for the effective application of control programmes on a case-specific basis [21]. However, in most of these studies only a small number of nematode species have been recognized and interspecies competitive effects, which may affect the worm numbers recovered, have not been sufficiently taken into consideration [20]. Moreover, several other epizootiological studies have not elucidated the seasonal pattern of *H. contortus* infections during the year, with the frequency and reliability of observations greatly varying. To overcome the aforementioned disadvantages, our study design included exclusively adult *H. contortus* collected from ruminants’ abomasum by the same researcher (K.V.A.) within a two-year period.

*Haemonchus contortus* is one of the two most pathogenic parasites (along with *Teladorsagia circumcincta*, Stadelmann, 1894) in the small ruminant industry, infecting the epithelium of abomasum [22]. It causes severe anaemia due to its blood-sucking activity, leading to significant monetary losses [23]. The investigation of the spatio-temporal spreading of haemonchosis and the update of its epizootiology are urgent matters for its sustainable control, considering, also, the constantly increasing anthelmintic resistance of *H. contortus*.

In Greece, the epizootiology of haemonchosis in sheep and goats has been reported by previous relevant studies, which were mainly based on the enumeration of L_3_ larvae in coprocultures [9,15,24,25,26]. According to the aforementioned data, haemonchosis in dairy small ruminants in Greece ranged from 22.4 to 32.0%. In our abattoir survey, the infection rate ranged from 32.0% for goats to 41.0% for sheep. The detection of mono-species *H. contortus* infection in Greek territory, which was recorded in our abattoir survey, has been previously confirmed by Arsenopoulos et al. [19]. Comparing the results of our study with past studies, it is noteworthy that there seems to be a progressive increase in the infection rate, possibly attributable to the observed anthelmintic resistance of strains (e.g., against pro/benzimidazoles) [27] and the climate change pattern (tropical–subtropical parasite), which also seems to be linked with the increased spatio-temporal distribution of the parasite [28,29].

Indeed, the extensive *H. contortus* distribution and the increased prevalence values in small ruminants, since the mid-2010s, suggest a wider geographical range where *H. contortus* is currently endemic, particularly in colder temperate climates of the northern hemisphere. Under these climatic conditions, the increased prevalence of haemonchosis has been linked to the increasing temperatures deriving from long-term climate changes [27,30,31,32].

In our study, a statistical analysis evaluated the effectiveness of pro/benzimidazoles, macrocyclic lactones, and their combination against *H. contortus* infection. More precisely, sheep treated with pro/benzimidazoles had the lowest likelihood to be infected by *H. contortus*, when compared to non-treated ones. Indeed, albendazole remains the most widely used benzimidazole in Greece, registered for per os administration (i.e., tablets or oral suspension), expressing a short-term efficacy against most internal parasites (i.e., nematodes, tapeworms, and liver flukes), shorter milk withdrawal period, and lower cost, compared to macrocyclic lactones [33]. On the other hand, macrocyclic lactones have a narrower antiparasitic spectrum against internal parasites, a longer milk withdrawal period (i.e., ivermectin), and a higher cost, when compared to pro/benzimidazoles. Eprinomectin, which is a macrocyclic lactone registered to overcome these limitations, combines a wide spectrum antiparasitic activity against GINs, lungworms, and some ectoparasites with a zero-day milk withdrawal period [25,26]. In our study, approximately 8% of the participated farms used eprinomectin in their applied antiparasitic protocols, with an increasing trend of its use.

We anticipated a reduced efficacy of pro/benzimidazoles against *H. contortus* since our previous field studies indicated that the extensive use of pro/benzimidazoles has enhanced the development of benzimidazole-resistant GINs, posing an emerging threat for the Greek sheep- and goat-farming sector [16,27]. These two latter studies examined adult female *H. contortus*, collected from both sheep and goats, to detect SNPs in the amino acid at position 200 of the gene encoding isotype-1 of β-tubulin; according to their findings, in the majority of the studied worms, homozygous alleles provoking benzimidazole resistance (i.e., Tyr/Tyr) were detected, while the homozygous alleles (TTC/TTC) at the same position and gene which encodes the susceptible amino acids Phe/Phe have not been detected [16,27]. This alarming situation has been further confirmed in many countries with a developed ruminant sector, compromising the efficacy of pro/benzimidazoles against GINs [34,35,36,37]. However, new information published by Babjak et al. [38] reported that the efficacy of pro/benzimidazoles in a benzimidazole-resistant population of *H. contortus* in both sheep and goats could be affected by the different proportion of homozygous- and heterozygous-resistant alleles, in total, and not exclusively based on the benzimidazole resistance associated with the isotype-1 β-tubulin gene codon 200 allele, justifying the results of the present study.

Contrary to sheep, the treatment with macrocyclic lactones in goats was followed by the lowest likelihood of *H. contortus* infection, in comparison with the non-treated ones. This is mainly attributed to differences in the gastrointestinal physiology of goats compared to sheep. Goats tend to have a faster ruminal turnover rate [39], which permit us to assume that the drugs administered via the oral route (i.e., pro/benzimidazoles) pass through the digestive system more quickly, compared to injectable or pour-on ones (i.e., macrocyclic lactones), explaining the better anthelmintic outcome against *H. contortus* in goats, in the present study. Moreover, the pour-on application of macrocyclic lactones such as eprinomectin in sheep is not an easy task in practice because it requires the partial removal of the fleece and direct contact of the applicator bottle spout on the skin over the backline [26]. This task is easier on the haircoat of the goats, possibly leading to a better absorption of the drug and, therefore, a better anthelmintic outcome.

It is noteworthy that newly published studies concluded that a similar or the same efficacy of pro/benzimidazoles is expected when mono-infection with *H. contortus* is suspected and when the same dose of benzimidazole is applied in both sheep and goats, despite their interspecific differences. This means that an underdosing should not be considered when goats are predominantly infected with *H. contortus* and are treated with an albendazole dose at 5 mg/kg of body weight (same as recommended for sheep), since the estimation of the representation of the resistant and susceptible parasites in the population is likely accurate [38].

The overall farming system and, particularly, grazing activity and practices contribute to the ruminants’ exposure to the L_3_ larvae of *H. contortus*. In Greece, two types of farming systems prevail, the intensive and the semi-intensive one [40]. In semi-intensive systems, feeding regimes are characterized by a combination of year-round grazing on pasturelands, usually close to the farm, and the supplementary feeding of concentrates and forages [17]. Under these systems, animals are often exposed to GINs. On the contrary, in intensive farming systems, sheep and goats remain permanently housed, with access (in some cases) to fenced yards; therefore, they have a lower exposure to GINs and, subsequently, *H. contortus* when reared under these systems [41]. This is consistent with our findings that revealed a higher likelihood of infection with *H. contortus* for sheep and goats living under semi-intensive systems compared to those living under intensive ones.

Host susceptibility/resistance to haemonchosis is known to be associated with the animal’s age [42]. Adult animals are more resistant to haemonchosis compared to young animals, due to a more efficient immune system response [42,43]. However, in our study, adult sheep were more likely to be infected with *H. contortus* compared to lambs. This is consistent with the fact that, in the studied farms, grazing was practiced exclusively in adult animals while lambs remained housed and, therefore, were not directly exposed to GIN.

Farm altitude has been reported as a potential risk factor for haemonchosis [44]. In that case, the differences between lowlands and highlands regarding the pedoclimatic conditions and particularly the ambient temperature seem to be related to the increased parasitic burden in lowlands, as it has been found that the development of the free larval stages of *H. contortus* requires high ambient temperatures [33]. This is in agreement with the findings of our study, where farms located at lowlands demonstrated a higher risk of haemonchosis. Another possible explanation for the aforementioned result is the different types of pasturelands and vegetation thereof in lowland and highland areas, which is associated with either the grazing or browsing activities demonstrated according to it; grazing activity facilitates the exposure of animals to *H. contortus* infections, contrary to browsing activity.

*Haemonchus contortus* is probably the most extensively studied nematode in small ruminants, due to its spreading, its importance, and its relevance with ecological factors that determine the viability of the eggs and its larval stages. Among them, seasonal changes of the weather play a critical role in *H. contortus* transmission and life cycle by affecting the survival and development of the free-living larval stages on pasture [45]. In general, *H. contortus* is regarded as being well-adapted to warm and humid climates [23], while it also poses a serious seasonal threat to ruminants living in temperate regions, such as Greece. This happens because of the increased ambient temperatures, which is the norm for several months during the year, allowing the development of the larval forms of the parasite, while the relatively shorter cold season results in the insufficient reduction in the parasitic burden in the pasturelands [33]. Most of the *H. contortus* eggs are dispersed during late winter and early spring. At this time, the parasitic burden in animals increases significantly. This increase is attributed to (i) the reactivation of the hypobiotic L_4_ larvae in ruminants’ abomasum, which results in a rise in adult worm populations and (ii) the periparturient rise in the egg output, which begins approximately two weeks before and lasts up to eight weeks after parturition. It was speculated that this rise was triggered by the immunosuppression caused by hormonal changes at the periparturient period, allowing a higher parasitic load [33,46], even though there was no clear association of a hormonal interference to the maintenance of this periparturient rise [47,48]. This increase in the parasitic burden on pastures contributes to the infection of animals with L_3_ larvae in late spring and early summer. Then, they develop into L_4_ larvae, which either cause the disease by feeding on blood or manifest the phenomenon of hypobiosis, re-infecting the pastures during the next productive season [33]. In our study, a statistically significant decrease in the likelihood of a goat to be infected by adult *H. contortus* was observed in spring and summer compared to winter.

In Greece, goats give birth mainly from January to March, the periparturition period at which most farmers choose to apply their anthelmintic treatments. The benefits of treating dams at parturition, when milk is consumed by the newborn kids or earlier during the last month of gestation, are the increased milk production and the avoidance of milk withdrawal periods. The grazing of the goats and, subsequently, their constant exposure to *H. contortus* initiate after the weaning of their kids and last approximately until one month before parturition (i.e., December). Therefore, it is obvious that, under these conditions, goats can be easily infected with *H. contortus* L_3_ larvae which, in turn, accumulate in the abomasum and evolve into adult helminths. This accumulative effect is more profound during winter months (i.e., several months post-anthelmintic treatment and constant grazing) and is enhanced by the strong selective pressure prevailing on pastures during summer months. Finally, this subsequent decrease in *H. contortus* burden may result from the development of host immunity, as a result of better nutrition (i.e., the presence of green vegetation) during spring and early summer compared to winter [9].

## 5. Conclusions

This study provides crucial insights into the epizootiology of *H. contortus* infection, revealing a higher prevalence of the infection in dairy small ruminants compared to previous epizootiological regional studies. Our findings highlighted that dairy sheep treated with pro/benzimidazoles and their lambs under two months old were significantly less likely to be infected by adult *H. contortus*. Similarly, the use of macrocyclic lactones, either alone or in combination with pro/benzimidazoles, were associated with a marked reduction in infection rates of dairy goats. Moreover, goat-raising farms experienced lower infection rates during the spring and summer months, a situation further confirmed at farms located over 300 m above sea level, underscoring the role of environmental factors in disease transmission. Finally, *H. contortus* infection was a minor problem in intensively managed dairy small ruminants. These findings emphasize the urgent need for targeted management practices and region-specific veterinary protocols to effectively reduce parasitic burdens. By addressing farm- and management-associated risk factors and implementing robust nematode control strategies, we can significantly mitigate the spread of *H. contortus* and curb the growing threat of anthelmintic resistance.

## Figures and Tables

**Figure 1 animals-15-00487-f001:**
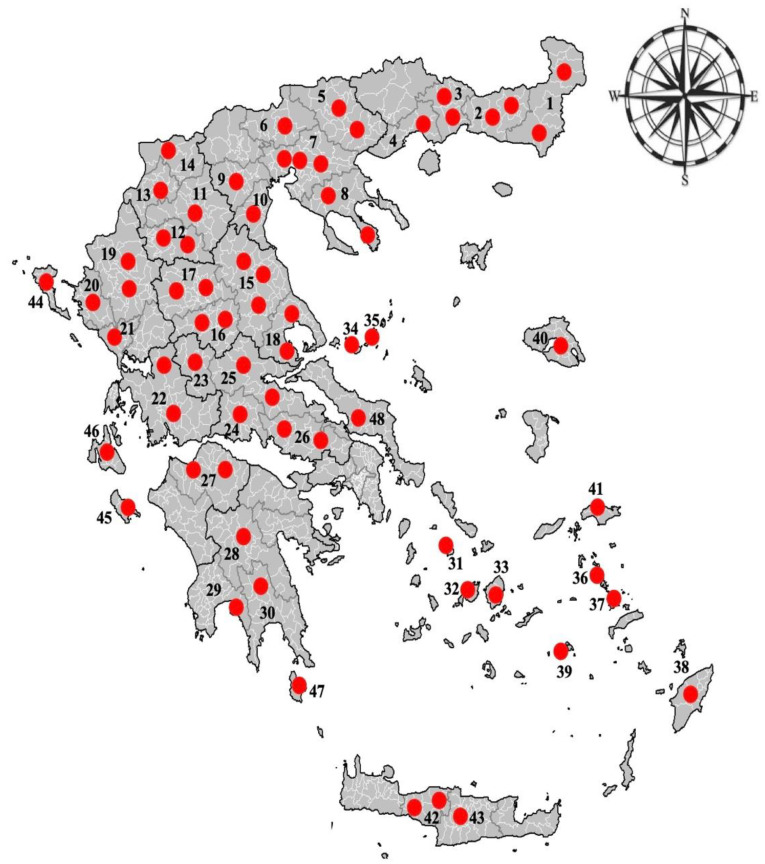
Location of the abattoirs around Greece, which were visited for the collection of sheep’s and goats’ abomasa. Numbers 1 to 48 represent different prefectures/islands included in the study.

**Figure 2 animals-15-00487-f002:**
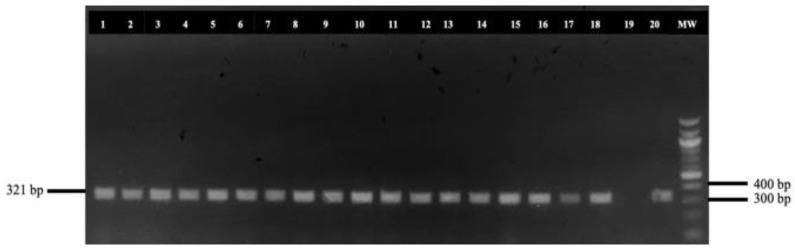
Detection of the amplified internal transcribed spacer 2 (ITS2) of *Haemonchus* spp. MW: 100 base pairs ladder; 1–18 and 20: amplified ITS2 sequences of collected *Haemonchus* spp.; and 19: negative control sample.

**Table 1 animals-15-00487-t001:** Categorized data included in the questionnaire.

1. Abattoir Information
**Location of the** **abattoir**
**Visit day**
**Season**
Spring
Summer
Autumn
Winter
2. Animals Information
**Age**
<2 months
2–15 months
>15 months
**Sex**
Male
Female
**Species**
Sheep
Goat
3. Farm Information
**Management system**
Semi-intensive
Intensive
**Altitude**
<300 m a.s.l.
>300 m a.s.l.
**Co-existence of sheep and goats**
Single-species farming
Mixed-species farming
**Anthelmintic treatment**
Exclusively pro/benzimidazoles
Exclusively macrocyclic lactones
Combination of pro/benzimidazoles
and macrocyclic lactones
No anthelmintic treatment

a.s.l.: above sea level.

**Table 2 animals-15-00487-t002:** Prevalence values (±95% Confidence Intervals) of *Haemonchus contortus* infection in sheep and goats in various prefectures/islands (based on the enumeration in Figure 1) of Greece.

Greece	Region	Prefecture/Island	Numbers from Figure 1	Sheep (%)	Goats (%)
**Continental**	Thrace	Evros	1	26.7 (10.9–52.0)	11.1 (2.0–43.5)
		Rodopi	2	18.8 (6.6–43.0)	25.0 (7.2–59.1)
		Xanthi	3	40.0 (19.8–64.3)	27.3 (9.8–56.6)
	Macedonia	Kavala	4	28.6 (11.7–54.7)	16.7 (4.7–44.8)
		Serres	5	42.9 (21.4–67.4)	13.3 (3.7–37.9)
		Kilkis	6	42.9 (21.4–67.4)	36.4 (15.2–64.6)
		Thessaloniki	7	43.8 (23.1–66.8)	33.3 (15.2–58.3)
		Chalkidiki	8	42.9 (21.4–67.4)	26.7 (10.9–52.0)
		Imathia	9	35.7 (16.3–61.2)	45.5 (21.3–72.0)
		Pieria	10	28.6 (11.7–54.7)	50.0 (18.8–81.2)
		Kozani	11	31.3 (14.2–55.6)	21.4 (7.6–47.6)
		Grevena	12	25.0 (10.2–49.5)	42.9 (15.8–75.0)
		Kastoria	13	29.4 (13.3–53.1)	25.0 (7.2–59.1)
		Florina	14	57.1 (32.6–78.6)	18.2 (5.1–47.7)
	Thessaly	Larissa	15	50.0 (29.0–71.0)	53.3 (30.1–75.2)
		Karditsa	16	31.6 (15.4–54.0)	47.1 (26.2–69.0)
		Trikala	17	38.9 (20.3–61.4)	36.8 (19.2–59.0)
		Magnesia	18	29.4 (13.3–53.1)	35.0 (18.1–56.7)
	Epirus	Ioannina	19	28.6 (11.7–54.7)	20.0 (5.7–51.0)
		Thesprotia	20	35.7 (16.3–61.2)	27.3 (9.8–56.7)
		Preveza	21	35.7 (16.3–61.2)	36.4 (15.2–64.6)
	Central Greece	Aitoloakarnania	22	42.9 (21.4–67.4)	37.5 (13.7–69.4)
		Evritania	23	50.0 (26.8–73.2)	42.9 (15.8–75.0)
		Fokida	24	35.7 (16.3–61.2)	60.0 (23.1–88.2)
		Fthiotida	25	42.9 (21.4–67.4)	25.0 (7.2–59.1)
		Viotia	26	50.0 (26.8–73.2)	37.5 (13.7–69.4)
	Pelopponnese	Achaia	27	57.1 (32.6–78.6)	28.6 (8.2–64.1)
		Arkadia	28	42.9 (21.4–67.4)	14.3 (2.6–51.3)
		Messinia	29	64.3 (38.8–83.7)	28.6 (8.2–64.1)
		Lakonia	30	71.4 (45.4–88.3)	14.3 (2.6–51.3)
**Insular**	Cyclades	Syros	31	83.3 (43.7–93.0)	33.3 (9.7–70.0)
		Paros	32	33.3 (9.7–70.0)	33.3 (9.7–70.0)
		Naxos	33	33.3 (9.7–70.0)	50.0 (18.8–81.2)
	Nothern Sporades	Skopelos	34	83.3 (43.7–97.0)	33.3 (9.7–70.0)
		Alonnisos	35	33.3 (9.7–70.0)	33.3 (9.7–70.0)
	Dodekanese	Leros	36	66.7 (30.0–90.3)	33.3 (9.7–70.0)
		Kalimnos	37	50.0 (18.8–81.2)	50.0 (18.8–81.2)
		Rhodes	38	33.3 (12.1–64.6)	33.3 (12.1–64.6)
		Astypalea	39	16.7 (3.0–56.4)	16.7 (3.0–56.4)
	Northern–Eastern Aegean	Lesvos	40	50.0 (18.8–81.2)	33.3 (12.1–64.6)
		Samos	41	16.7 (3.0–56.4)	16.7 (3.0–56.4)
	Crete	Rethymno	42	83.3 (43.7–97.0)	50.0 (18.8–81.2)
		Heraklion	43	83.3 (43.7–97.0)	50.0 (18.8–81.2)
	Ionian	Corfu	44	20.0 (5.7–51.0)	16.7 (3.0–56.4)
		Zakynthos	45	33.3 (9.7–70.0)	33.3 (9.7–70.0)
		Kefallonia	46	50.0 (18.8–81.2)	50.0 (18.8–81.2)
		Kythira	47	16.7 (3.0–56.4)	16.7 (3.0–56.4)
	Central Greece	Evia	48	53.3 (30.1–75.2)	33.3 (9.7–70.0)

**Table 3 animals-15-00487-t003:** Numbers and percentages of sheep and goats infected by *Haemonchus contortus* per farm- and host-related risk factor (i.e., co-existence of sheep and goats, anthelmintic treatment, age, farm location altitude, management system, sex, and season).

	Sheep *	%	Goats *	%
**Co-existence of sheep and goats**				
Single-species farming	162	68.9	96	69.6
Mixed-species farming	73	31.1	42	30.4
**Anthelmintic treatment**				
Exclusively pro/benzimidazoles	125	53.2	85	61.6
Exclusively macrocyclic lactones	27	11.5	11	08.0
Combination of pro/benzimidazoles and macrocyclic lactones	27	11.5	13	09.4
No anthelmintic treatment	56	23.8	29	21.0
**Age**				
<2 months	42	17.9	56	40.6
2–15 months	113	48.1	58	42.0
>15 months	80	34.0	24	17.4
**Altitude**				
<300 m a.s.l.	160	68.1	118	85.5
>300 m a.s.l.	75	31.9	20	14.5
**Management system**				
Semi-intensive	193	82.1	130	94.2
Intensive	42	17.9	8	05.8
**Sex**				
Male	106	45.1	61	44.2
Female	129	54.9	77	55.8
**Season**				
Spring	36	15.3	34	24.6
Summer	85	36.1	37	26.8
Autumn	59	25.2	41	29.8
Winter	55	23.4	26	18.8

a.s.l.: above sea level. * The total numbers of sheep and goats with *H. contortus* infection are 235 and 138, respectively.

**Table 4 animals-15-00487-t004:** Beta-coefficients (±standard errors), Wald test, *p*-values, and odds ratios with 95% confidence intervals of the risk factors of sheep infected by *Haemonchus contortus*.

	B	S.E.	Wald	*p*-Value	Odds Ratio	95% C.I. for Odds Ratio	
						Lower	Upper
Mixed-species farming	−0.12	0.214	0.30	0.584	0.89	0.59	1.35
Single-species farming	*Ref.*						
Pro/benzimidazoles	−2.19	0.381	33.06	0.000	0.11	0.05	0.24
Macrocyclic lactones	−2.03	0.444	20.83	0.000	0.13	0.06	0.32
Pro/benzimidazoles and macrocyclic lactones	−2.14	0.440	23.64	0.000	0.12	0.05	0.28
No anthelmintic treatment	*Ref.*						
<2 months	−1.61	0.279	33.15	0.000	0.20	0.12	0.35
2–15 months	−0.19	0.247	0.59	0.441	0.83	0.51	1.34
>15 months					*Ref.*		
<300 m a.s.l.	0.04	0.209	0.04	0.843	1.04	0.69	1.57
>300 m a.s.l.	*Ref.*						
Semi-intensive	1.00	0.234	18.45	0.000	2.73	1.73	4.32
Intensive	*Ref.*						
Male	0.05	0.196	0.07	0.797	1.05	0.72	1.55
Female	*Ref.*						
Spring	0.24	0.337	0.50	0.479	1.27	0.66	2.46
Summer	0.26	0.271	0.89	0.346	1.29	0.76	2.20
Autumn	−0.53	0.293	3.21	0.073	0.59	0.33	1.05
Winter	*Ref.*						
Constant	1.44	0.499	8.37	0.004	4.23		

Β: b-coefficient, S.E.: standard error, C.I.: confidence interval, a.s.l.: above sea level, *Ref.*: Reference category.

**Table 5 animals-15-00487-t005:** Beta-coefficients (±standard errors), Wald test, *p*-values, and odds ratios with 95% confidence intervals of the risk factors of goats infected by *Haemonchus contortus*.

	B	S.E.	Wald	*p*-Value	Odds Ratio	95% C.I. for Odds Ratio	
						Lower	Upper
Mixed-species farming	0.05	0.250	0.04	0.835	1.05	0.65	1.72
Single-species farming					*Ref.*		
Pro/benzimidazoles	−3.95	0.811	23.69	0.000	0.02	0.00	0.10
Macrocyclic lactones	−4.40	0.879	25.01	0.000	0.01	0.00	0.07
Pro/benzimidazoles and macrocyclic lactones	−4.30	0.858	25.01	0.000	0.01	0.00	0.07
No anthelmintic treatment					*Ref.*		
<2 months	0.23	0.310	0.53	0.467	1.25	0.68	2.30
2–15 months	0.22	0.313	0.47	0.491	1.24	0.67	2.29
>15 months					*Ref.*		
<300 m a.s.l.	0.97	0.309	9.75	0.002	2.63	1.43	4.82
>300 m a.s.l.					*Ref.*		
Semi-intensive	1.20	0.545	4.81	0.028	3.30	1.14	9.62
Intensive					*Ref.*		
Male	−0.02	0.235	0.01	0.933	0.98	0.62	1.55
Female					*Ref.*		
Spring	−0.96	0.362	6.98	0.008	0.38	0.19	0.78
Summer	−0.94	0.386	5.87	0.015	0.39	0.18	0.84
Autumn	−0.55	0.350	2.43	0.119	0.58	0.29	1.15
Winter					*Ref.*		
Constant	1.63	0.980	2.76	0.097	5.09		

Β: b-coefficient, S.E.: standard error, C.I.: confidence interval, a.s.l.: above sea level, *Ref.*: Reference category.

## Data Availability

The datasets used and/or analyzed during the current study are available from the corresponding author upon reasonable request.

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
