# Peer review of "Abattoir Countrywide Survey of Dairy Small Ruminants’ Haemonchosis in Greece and Associated Risk Factors"

_animals, 2025, doi:10.3390/ani15040487_

Round 1
Reviewer 1 Report
Comments and Suggestions for Authors
Please see attachment.

Reviewer 2 Report
Comments and Suggestions for Authors
Review
I am pleased to have the opportunity to review a paper on such an important topic and a currently recognized challenge posed by haemonchosis to livestock production in Europe and worldwide. First of all, I would like to commend the authors.
Abstract
It might be better not to emphasize objectives, methods, and results in the abstract. I would also suggest adding a concluding sentence to make the abstract more concise.
Background
I suggest enriching the introduction with information about risk factors and their impact on ruminants, considering these factors were evaluated in the study.
Methods
I propose presenting the data in the "Data Collection" section in a slightly different way to enhance clarity. Perhaps using a table with explanations, as it was done in the Results section.
Line 141: It would be good to include where the samples were delivered.
Results
The title should be "Molecular identification of Haemonchus contortus" since the species has already been confirmed. Otherwise, this might confuse the reader.
Line 163: Consider using the title "Prevalence of H. contortus infection."
Table 3: Did you consider the number of anthelmintic treatments per year?
Discussion
Line 233: Could you clarify what traditional livestock management in Greece entails?
Lines 245-253: Could you provide a reference for this part of the discussion?
Line 272: It should state "Haemonchus contortus" instead of "H. contortus"
Lines 331-336: Please provide the appropriate reference.
Line 374: “Haemonchus contortus"
Reviewer 3 Report
Comments and Suggestions for Authors
The paper is devoted to a very important topic of parasitological research on zoonotic parasites of Small Ruminants in Greece Ruminants. This is an excellent manuscript where the authors have perfectly summarized the research necessity and findings in good English. The data on distribution on nematodes Haemonchus contortus in Greece are original and essential. Authors have studied a large amount of material over a considerable area. Researchers also not only studied infection of dairy sheep and goats in Greece by Haemonchus contortus using an integrative approach, but also evaluated host-related risk factors for infection.
Thus, the manuscript is suitable for publication in Animals after the several corrections mentioned in the text file and below:
1. Please add Author and Year for every species at first mention in the text
2. Line 60: Write the genus name in full.
3. Please check the spelling of “ITS2” in Lines 145 and following.
4. Line 149: Table 1 can be deleted, and write the name and sequence of the primers directly in the text.
5. Line 157-161: I recommend that you also submit the sequences you received to GenBank, at least 1-2 from each sampling place.
6. Table 2: Write the head of the “Numbers” column as “Numbers from Figure 1”. That's more understandable.

Reviewer 4 Report
Comments and Suggestions for Authors
see the pdf sent to your the editor

Round 2
Reviewer 1 Report
Comments and Suggestions for Authors
Thank you for responding to my editorial comments. I am satisfied that you have adequately addressed all of the comments with one exception.
Reviewer Comment - Original Line 114: If I understand correctly, you did not test for interaction among these effects, so it is not possible to be confident in the causative role of each effect? For example, was drench usage different among farms in the two different altitude groupings? Might the significance of this effect in Table 5 be due to factors confounded with altitude?
Authors answer: During the initial stages of analyses full factorial models and potential interactions were tested to check for significant confounders. No significant effects of these interactions were found and hence, ‘main-effects’ models were used instead.
I suggest the authors consider including their answer they provided above in the relevant statistical section of the manuscript to give greater confidence to readers in the results of their work.
